# The Effect of Copper on the Microstructure, Wear and Corrosion Resistance of CoCrCuFeNi High-Entropy Alloys Manufactured by Powder Metallurgy

**DOI:** 10.3390/ma16031178

**Published:** 2023-01-30

**Authors:** Samat Mukanov, Pavel Loginov, Alexander Fedotov, Marina Bychkova, Maria Antonyuk, Evgeny Levashov

**Affiliations:** Department of Powder Metallurgy and Functional Coatings, National University of Science and Technology «MISIS», Leninskiy Prospekt 4, 119049 Moscow, Russia

**Keywords:** high-entropy alloy, powder metallurgy, structure, phase composition, mechanical properties, friction, wear, corrosion

## Abstract

This paper focuses on the microstructure, phase composition, mechanical, tribological and corrosion properties of high-entropy alloys (HEAs) in the CoCrCuFeNi system depending on copper content, which was varied from 0 to 20 at. % with an increment of 5%. CoCrCuFeNi alloys were manufactured by powder metallurgy methods: mechanical alloying and hot pressing of element mixtures. The solubility limit of copper in CoCrFeNi solid solution was found to be 9 at. %. Segregation of irregularly shaped copper grains sized 1–30 μm is observed at concentrations above this solubility limit. As copper concentration increases, the phase composition of CoCrCuFeNi alloys changes from the single phase based on FCC1 solid solution (Cu = 0–5 at. %) to the dual-phase FCC1 + FCC2 alloy (Cu = 10–20 at. %), where FCC1 is the main phase and FCC2 is the secondary copper-rich phase. Tribological tests have shown that doping the CoCrFeNi alloy with copper increased wear resistance by 23% due to solid solution hardening. As copper content rises above 20%, the content of the secondary FCC2 phase increases, while wear resistance and alloy hardness decline. An analysis of wear tracks and wear products has shown that abrasion of CoCrCuFeNi alloys occurs via the abrasive-oxidative wear mechanism. The corrosion tests of CoCrCuFeNi HEAs in 3.5% NaCl solution had demonstrated that doping the alloy with copper at low concentrations (5–10%) leads to decreasing of corrosion resistance, possibly due to the formation of undesirable oxide Cu_2_O along with protective Cr_2_O_3_. At high copper concentrations (15–20%) galvanic corrosion is suppressed due to coarsening of FCC2 grains and thus decreasing the specific contact surface area between the cathode (FCC2) and the anode (FCC1).

## 1. Introduction

Developing novel alloys simultaneously characterized by excellent mechanical, tribological, and corrosion properties is one of relevant challenges in materials science. There exist many approaches to solving this problem: macro- and micro-doping of alloys; formation of special types of the structure (duplex, ultrafine, nanocrystalline, etc.); incorporating reinforcing nanoparticles or their formation during consolidation or heat treatment of alloys; implementing the TWIP and TRIP effects to achieve a balance between strength and ductility, etc. These approaches have been successfully used over the past 20 years to design novel materials under the concept of high-entropy alloys (HEAs), which was proposed by Cantor et al. [1] and Yeh et al. [2] in 2004. This class of materials includes alloys consisting of at least five components with their concentrations ranging from 5 to 35 at. %. A distinctive feature of HEAs is that a thermodynamically stable substitutional solid solution with the initial fcc, bcc, or hcp lattice [3] is formed due to the high entropy of mixing (>1.5 R [4]). This approach allows us to design new alloys with excellent combination of strength, ductility, hardness and wear resistance [5,6]. The presence of a high concentration of uniformly distributed passive oxide-forming elements such as Cr ensures outstanding corrosion resistance [7]. These features of HEAs provide their superiority over traditional steels or iron-based alloys for multi-purpose materials [8,9,10].

CoCrFeNi alloys are common HEAs. These HEAs are promising to be used as materials for manufacturing press molds, components of magnetic systems, catalysts, etc. [11]. One of important features of most HEAs belonging to this family is their high wear resistance that equals or even exceeds that of commercial-grade engineering steels, and steel 304 L in particular [11].

Improvement of the known CoCrFeNi-based HEAs is performed in compliance with the strategy for designing alloys exhibiting an effect of solid solution and dispersion strengthening by doping with various elements [12,13,14]. Al, Mn, V, and Nb are the most common modifying agents for CoCrFeNi-based HEAs. Thus, Yang et al. [14] found that doping the CoCrFeNi alloy with Al results in rearrangement of the initial fcc crystal lattice into the non-ordered (A2) or ordered (B2) bcc structure, which is accompanied by an increase in hardness from 1.83 to 5.6 GPa. This effect was proved in refs. [11,15,16,17]. Joseph et al. [17] carried out comparative wear resistant tests of AlCoCrFeNi HEA, stainless steel (AISI 304), and Inconel 718 superalloy in a broad temperature range up to 900 °C. Hardness of the equiatomic AlFeCoNiCr alloy (630 HV) was approximately 2.5-fold higher than that of steel (245 HV) and 1.5-fold higher than that of nickel superalloy (425 HV). The presence of the bcc phase was found to significantly increase hardness and wear resistance, while simultaneously reducing ductility [18]. Zhang et al. [19] reported the results of a comparative study of the effect of V and Mn in CoCrFeNi HEA on mechanical and tribological properties. Doping with vanadium also resulted in significant alloy reinforcement and improved wear resistance threefold compared to that of the CoCrFeNiMn HEA due to precipitation of the σ-CrV solid phase. Wear resistance and tribological properties of CoCrFeNi-based HEAs can also be improved by doping them with ductile metals such as copper [20,21,22,23,24]. It seems to be a promising candidate for improving the CoCrFeNi-based HEA because its physical and chemical properties are similar to those of the matrix. Nevertheless, the available data on mechanical and tribological properties of CoCrCuFeNi-based HEAs are controversial and fail to provide an unambiguous answer to the question whether copper doping is beneficial. Verma et al. [20] studied the effect of doping with 5, 9, 13, 17, and 20 at. % Cu on the microstructure and wear resistance of the CoCrFeNi alloy obtained by arc remelting in an argon medium. The increased copper content resulted in grain size reduction. In alloys with Cu content >2 at. %, the secondary phase consisting of pure copper precipitates was observed in the interdendritic region along with the fcc solid-solution phase. As copper concentration increased, hardness of the alloy rose from 136 to 169 HV. Tribological tests of HEAs at 25 and 600 °C showed that doping with up to 20 at. % copper increased wear resistance by 26% and 48%, respectively. At elevated temperatures, there was a self-lubricating effect caused by formation of CuO oxide film. Contradictory data regarding the effect of copper were obtained in other studies [21,22,23]. Since copper has a positive enthalpy of mixing [24] with Fe-Co-Ni-Cr alloy elements (13; 6; 4; and 12 J/mol, respectively), copper atoms are segregated in the interdendritic space and/or at grain boundaries during crystallization. Therefore, such pure copper precipitates can adversely affect hardness, corrosion, and wear resistance of multicomponent alloys.

One of the possible ways to improve the mechanical, tribological properties and corrosion resistance of CoCrFeNi-based HEAs is to apply powder metallurgy techniques. Currently the most commonly used methods for their production (arc melting, casting, additive manufacturing and others) involve the formation of a liquid phase, resulting in inhomogeneity, dendritic segregation and formation of eutectic-type structures [25]. Mechanical alloying of powder mixtures followed by hot pressing or spark plasma sintering allows obtaining materials with fine structure and uniform distribution of the elements, despite the limitations or even complete absence of their mutual solubility. This study aimed to investigate the effect of copper on the microstructure, mechanical, tribological, and corrosion properties of CoCrCuxFeNi HEAs fabricated by powder metallurgy, which involved high-energy mechanical alloying of element mixtures followed by solid-phase hot pressing. This technology allows one to produce homogeneous poreless materials from single-phase supersaturated solid solutions. This fact makes these fine-grained alloys differ from cast coarse-grained alloys with a dendritic structure. This type of structure is undesirable if high-level mechanical properties, corrosion and wear resistance need to be achieved.

## 2. Materials and Methods

Carbonyl iron and nickel powders, cobalt and chromium powders obtained by reduction, and electrolytic copper powder were used in this study. The equiatomic CoCrCuFeNi alloy was chosen as the basic composition and was subsequently doped with 5, 10, 15, and 20 at. % copper. The samples are designated as CoCrCu_0.25_FeNi, CoCrCu_0.5_FeNi, CoCrCu_0.75_FeNi, and CoCrCuFeNi.

Powder mixtures were fabricated by high-energy mechanical alloying (MA) on an Activator-2 sl planetary ball mill (PBM) (OJSC Chemical Machine Building Plant, Novosibirsk, Russia) in the following mode: rotational speed of jars, 694 rpm; duration, 30 min; grinding bodies (steel balls)-to-powder weight ratio, 15:1. The jars were filled with argon to prevent powder oxidation during the treatment.

Cylinder-shaped compact samples 50 mm in diameter and 4 mm high were fabricated by vacuum hot pressing (HP) on a DSP-515 SA setup (Dr. Fritsch, Fellbach, Germany) at temperature of 950 °C, pressure of 35 MPa, and isobaric exposure of 3 min.

X-ray diffraction analysis (XRD) of powder mixtures and compact samples was carried out on an automated DRON 4-07 X-ray diffractometer using monochromatic Cu-Kα (for powders) and Co-Kα (for compact samples) radiation in the Bragg–Brentano geometry. The lattice constants were measured with a relative error ∆a/a = 10^−4^ nm. The microstructure and composition of HEAs were examined by scanning electron microscopy (SEM) on a S-3400N electron microscope (Hitachi, Tokyo, Japan). The fine structure was studied using a JEM-2100 transmission electron microscope (Jeol, Tokyo, Japan).

Tribological tests were carried out with a high-precision tribometer (CSM Instruments, Peseux, Switzerland) according to ASTM G 99-17 and DIN 50,324 standards using the “pin-on-disk” scheme upon reciprocating motion. Balls (diameter, 3 mm) made of sintered silicon nitride Si_3_N_4_ were used as counterbodies. Test loading was 2 N; linear speed was 5 cm/s. Wear track length was 6 mm for the total travel distance of 4000 cycles. Fractographic examination that involved measurements of the abrasive groove profile and diameter of the wear spot of the counterbody was carried out using a WYKO NT1100 optical profilometer (Veeco, Plainview, NY, USA) and an AXIOVERT CA25 inverted microscope (Karl Zeiss, Jena, Germany).

Hardness was measured on a Vickers HVS-50 hardness tester (Time Group Inc., Beijing, China) at a load of 10 N. The elastic modulus (E) was measured on a Nano-Hardness Tester (CSM Instruments, Peseux, Switzerland) using a Berkovich diamond indenter at a load of 8 mN. The corrosion resistance tests for HEAs were carried out in 3.5% NaCl solution in a three-electrode cell connected to a VoltaLab 50 potentiostat (Radiometer Analytical, Villeurbanne, France) at room temperature.

## 3. Results

### 3.1. Preparing Powder Mixtures of CoCrCuFeNi HEAs

One of the objectives of this study was to produce CoCrCu_x_FeNi HEAs having a homogeneous structure and characterized by complete mutual dissolution of the components. This is not always feasible in the Co-Cr-Cu-Fe-Ni system because of the low solubility of copper in Fe, Co, and Cr, as well as the presence of the eutectics in the corresponding phase diagram. As a result, most CoCrCu_x_FeNi cast alloys have a coarse dendritic structure with structural components varying significantly in terms of their chemical composition and mechanical properties (the primary crystals based on (Fe)_CoNiCr_ solid solution and the copper-based eutectics) [26,27,28,29,30].

Single-phase CoCrCu_x_FeNi precursor powders were produced by MA to achieve uniform distribution of elements (Figure 1). The four-component CoCrFeNi powders were a solid solution having a fcc crystal lattice and lattice constant of 0.3583 nm (designated as FCC). Doping with copper at concentration of 5–20 at. % did not give rise to other phases. Peak positions in the XRD patterns of CoCrCu_x_FeNi powders (peaks shifted to smaller 2θ angles) suggested that copper dissolution in the matrix increased the lattice period to 0.3609 nm (at 20 at. % Cu content).

The homogeneous structure was formed via the known mechanisms of mechanical alloying when high-intensity plastic deformation and friction welding of ductile metallic components took place [30,31].

The CoCrCuFeNi powder mixtures subjected to long-term treatment in the PBM were characterized by uniform distribution of elements (Figure 2).

### 3.2. The Structure of CoCrCuFeNi Compact Samples

Compact samples were prepared from MA powder mixtures characterized by different copper contents. Figure 3 shows the characteristic microstructures of CoCrCuFeNi HEAs with copper content of 0, 5, 10, 15, and 20 at. %. The basic CoCrFeNi alloy had a homogeneous structure, which indicates that the alloy contained a single phase. A single-phase structure was also observed when the alloy was doped with 5 at. % Cu, which indicated that copper was completely dissolved in the fcc solid solution. As copper content was increased to 10 at. % and 20 at. %, isostructural Cu-rich clusters were precipitated (their content being proportional to copper concentration).

The chemical composition of the respective structural component was studied by energy-dispersive X-ray spectroscopy (EDX) in order to assess the solubility limit of copper in the matrix phase. The samples doped with 10–20% copper were characterized by an identical copper content in this phase (9 at. %). The constant chemical composition of the matrix after doping with ≥10 at. % Cu presumably indicates that the solubility limit has been reached. These results agree with the data reported by Shkodich et al. [32].

Figure 4 shows the XRD patterns of hot-pressed CoCrCu_x_FeNi samples indicating that the basic CoCrFeNi alloy contained only the fcc solid solution with lattice constant a = 0.35752 nm (hereinafter FCC1). This fact demonstrates that phase transformations did not occur as powder mixtures were heated. In accordance with the views on copper solubility in the fcc lattice, phase segregation related to the positive enthalpy of mixing for the elements of the CoCrFeNi alloy [24,33,34] is observed at Cu contents >2 at. % [21]. However, the experimental data indicate that mechanical alloying and hot pressing increased the solubility limit of copper to 9 at. %. In particular, when doping the alloy with 5 at. % Cu, copper was completely dissolved in the fcc matrix solid solution, thus slightly increasing the lattice period (a = 0.35753 nm). XRD peak splitting was observed as copper content was increased above 10 at. %. It indicates that the secondary phase, hereinafter FCC2, was precipitated (lattice constants a = 0.36325 nm, 0.36111 nm and 0.35963 nm) for the CoCrCu_0.5_FeNi, CoCrCu_0.75_FeNi, and CoCrCuFeNi samples, respectively, being consistent with the literature data [35,36,37]. The intensity of XRD lines corresponding to the secondary FCC2 phase rose with increasing copper content in the HEA. The content of the FCC2 phase in the equiatomic CoCrCuFeNi alloy reached 15 wt.%. The average chemical composition of the FCC2 phase was identified by EDX: Cu, 90 at. %; Ni, 8 at. %; Fe, Co, and Cr, 2 at. % each.

The main reason for the formation of the FCC2 phase is extremely low mutual solubility of Cu with Co, Cr and Fe. Application of high energy processes, such as MA, for powder production allows us to obtain a solid solution with higher Cu concentration than is thermodynamically permissible. Thus, equiatomic MA powder mixture CoCrCuFeNi, obtained in our work, initially consists of a metastable phase, which is exposed to diffusion activated spinodal decomposition at hot pressing. The higher the amount of Cu is in CoCrCu_x_FeNi alloys, the higher is the concentration of FCC2 phase forms in the alloy.

### 3.3. Tribological and Physicomechanical Properties

Table 1 summarizes the results of tribological tests of the fabricated HEA samples with different Cu contents. Doping with copper in the specified range of concentrations and segregation of the secondary phase FCC2 did not noticeably change the coefficient of friction (Figure 5a). During the initial running-in of the friction couple at 250–360 cycles, the coefficient of friction decreased to 0.4 and subsequently increased to 0.7. The equiatomic CoCrCuFeNi alloy was characterized by a longer running-in period.

The wear track profiles of the CoCrCuFeNi alloy are shown in Figure 5b. One can see that doping with copper noticeably increased the wear resistance of CoCrFeNi (Table 1). There was an extremal dependence of reduced wear on copper concentration with a minimum at 5 at. % of Cu. Wear resistance of the CoCrCu_0.25_FeNi alloy increased due to solid solution hardening caused by crystal lattice distortion of the (Fe)_Co,Ni,Cr_ solid solution. The CoCrCu_0.5_FeNi alloy was also characterized by high wear resistance. In this case, wear resistance was affected by two competing processes: improvement of mechanical properties of the solid solution FCC1 due to copper saturation up to the solubility limit and segregation of the copper-based phase FCC2 characterized by low hardness. The first process was probably predominant due to the low content of the secondary phase FCC2 (5 wt.%). Wear resistance of HEA declined with increasing copper concentration. The highest values of reduced wear were observed for the CoCrCu_0.75_FeNi and CoCrCuFeNi alloys (8.88 and 10.54 × 10^−5^·mm^3^/N·m, respectively), which was probably caused by increasing concentration of the soft FCC2 phase.

Figure 6 shows the results of studying the effect of copper concentration on hardness, elastic modulus, and wear resistance of HEAs. One can see that hardness values are correlated rather tightly with wear resistance. The function showing the dependence of the hardness of the CoCrCu_x_FeNi HEA on copper concentration has a maximum at 10 at. % (374 HV). The increased hardness in the concentration range of 0–10 at. % was associated with solid-solution hardening of the main FCC1 phase by copper. The maximum mechanical properties (hardness and elastic modulus) of the FCC1 phase were attained at copper concentrations higher than its solubility limit (9 at. %). The concentration of Cu-rich FCC2 phase in CoCrCu_0.75_FeNi and CoCrCuFeNi alloys increases, which is detrimental for hardness and wear resistance. 

Figure 7 shows the SEM images of the wear tracks after the tribological tests. Grooves oriented along the direction of counterbody motion and showing signs of abrasive wear of the friction couple were detected on all the samples in the tribocontact zone (Figure 5 and Figure 7). In all the HEAs, the wear track surface was characterized by high content of regions with dark BSE contrast. EDX of these areas (Figure 8) shows the high oxygen concentration in the dark regions of abrasive grooves, which may be indicative of oxidation processes and, therefore, predominance of the oxidative wear mechanism. The partial formation of the oxide layer as a result of frictional heating is associated with the fact that Fe, Ni, Cr, and Cu are prone to forming rapidly growing oxides.

The cracks in the oxidized areas of wear tracks and partial delamination of the tribolayer are possibly related to the fact that the oxides formed during tests are more brittle and degrade more easily under the applied testing load. The difference in the coefficient of thermal expansion [34] compared to that of the initial FCC solid solution can be an additional factor contributing to delamination of the oxidized region. After local delamination of the oxidized regions of the wear tracks, the metal surface in the tribocontact zone can be reoxidized [38], which temporarily raises the coefficient of friction, thus increasing the wear of the material [14].

A detailed study of the wear products was performed by transmission electron microscopy (TEM) and energy-dispersive X-ray spectroscopy (EDXS). The tribolayer consisted of a combination of highly dispersed particles with a cloudy shape and an amorphous structure (Figure 9). No selective oxidation of any component of the CoCrCuFeNi alloy was observed during the tribolayer formation. EDXS revealed that the tribolayer also contained 5 at. % silicon, which had initially been a component of the counterbody. Silicon was uniformly distributed in the tribolayer particles along with HEA components and oxygen (Figure 9). Since no Si_3_N_4_ phase grains have been detected among the wear products, one can infer that chemical interaction between the CoCrCuFeNi HEA and the counterbody material takes place during tribological tests, followed by silicon dissolution.

### 3.4. Corrosion Resistance of the HEA (Corrosion Analysis)

Figure 10 shows the polarization curves of CoCrCu_x_FeNi compact samples in 3.5% NaCl solution. The corrosion potential (*E*) and corrosion current density (*I*) were determined using the Tafel extrapolation method. The CoCrFeNi alloy exhibited high corrosion resistance, which can be seen from the low corrosion current *I* (0.4 mA·cm^−2^). The shape of the polarization curve at high E values indicates that the alloy was resistant to pitting corrosion due to formation of a stable passivating film based on Cr_2_O_3_ chromium oxide [36]. Doping the HEA with copper reduced the corrosion resistance in the case when this element was completely dissolved in the FCC solid solution matrix. This is indicated by the maximum corrosion current density (*I* = 3.38 mA·cm^−2^), which was directly proportional to the corrosion rate according to the Faraday’s law [39]. In this case, the undesirable oxide Cu_2_O deteriorating the protective properties of the passivation layer was formed on the sample surface along with Cr_2_O_3_ [40].

At 10–20 at. % copper concentration, the corrosion current density decreased monotonically from 2.24 to 0.4 mA·cm^−2^, being indicative of increasing corrosion resistance. A typical feature of the structure of the resulting HEAs was that it contained the secondary Cu-based phase. The heterogenous structure ensured conditions for galvanic corrosion between the structural components acting as a cathode (FCC2) and an anode (FCC) [41]. The FCC2 grains became coarser as copper concentration increased, thus reducing the specific contact surface area between the cathode and the anode and increasing corrosion resistance.

## 4. Conclusions

Compact samples of CoCrCu_x_FeNi HEA with copper content ranging from 0 to 20 at. % were fabricated using the powder metallurgy technology that combined high-energy mechanical alloying and hot pressing. A single-phase solid solution was formed at copper concentrations up to 5 at. %, while at copper concentrations of 10–20 at. %, the structure was dual-phase. The excess phase having a fcc lattice precipitated during hot pressing consisted of 90 at. % copper and caused disordering of the HEA.The CoCrCu_0.25_FeNi alloy exhibited the maximum wear resistance upon friction in a couple with a counterbody made of sintered Si_3_N_4_. Doping with copper reduced the specific wear by 23% (from 6.68 to 5.17 10^−5^·mm^3^/N·m). The increase in wear resistance was attributed to solid solution hardening caused by distortion of the crystal lattice of the (Fe)_Co,Ni,Cr_ solid solution. Wear resistance declined with increasing copper concentration due to segregation of the excess FCC2 phase and decreasing hardness.Wear of the resulting CoCrCu_x_FeNi HEAs occurred via the abrasive-oxidation mechanism. The wear products were amorphous particles consisting of the alloy components with uniformly distributed oxygen and silicon elements.The tests of the CoCrCu_x_FeNi HEAs in 3.5% NaCl solution revealed that formation of the duplex structure FCC1 + FCC2 had a beneficial effect on corrosion resistance compared to the single-phase HEA with composition CoCrCu_0.25_FeNi.

## Figures and Tables

**Figure 1 materials-16-01178-f001:**
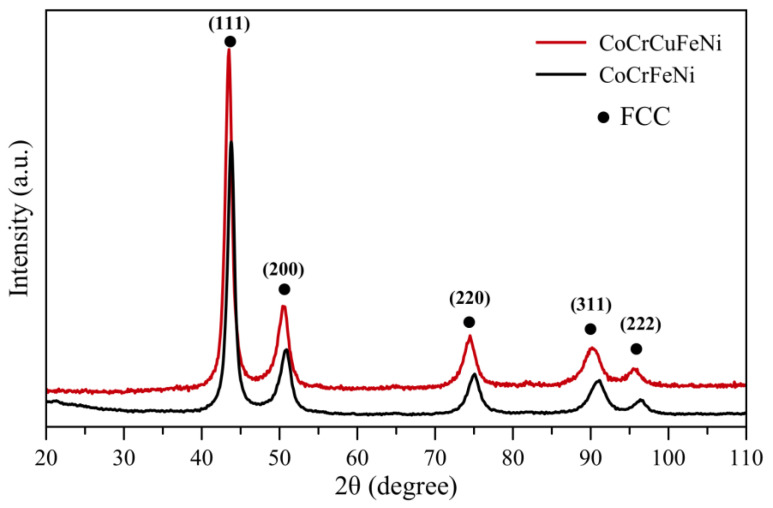
XRD patterns of MA CoCrFeNi and CoCrCuFeNi powder mixtures.

**Figure 2 materials-16-01178-f002:**
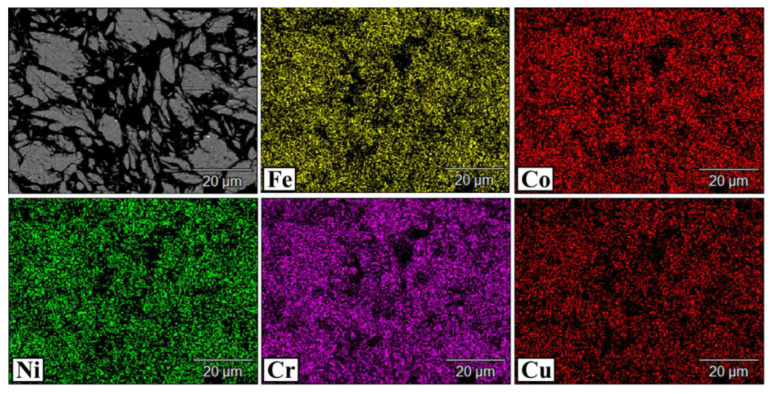
Element distribution map in the CoCrCuFeNi powder subjected to MA.

**Figure 3 materials-16-01178-f003:**
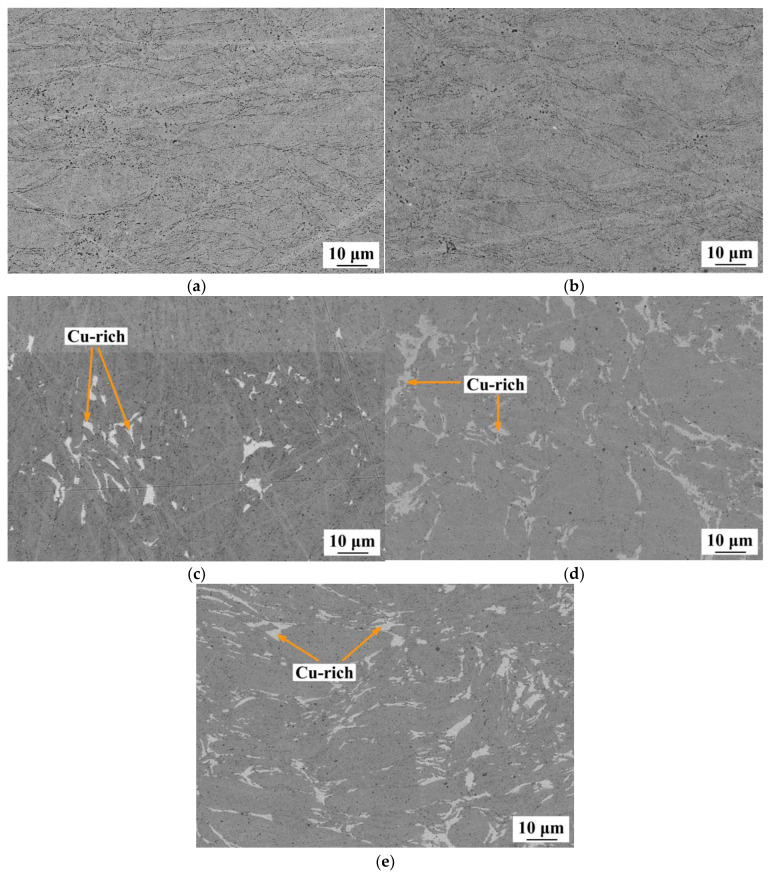
SEM images of the microstructure of the hot-pressed samples: (**a**) CoCrFeNi; (**b**) CoCrCu_0.25_FeNi; (**c**) CoCrCu_0.5_FeNi; (**d**) CoCrCu_0.75_FeNi; and (**e**) CoCrCuFeNi.

**Figure 4 materials-16-01178-f004:**
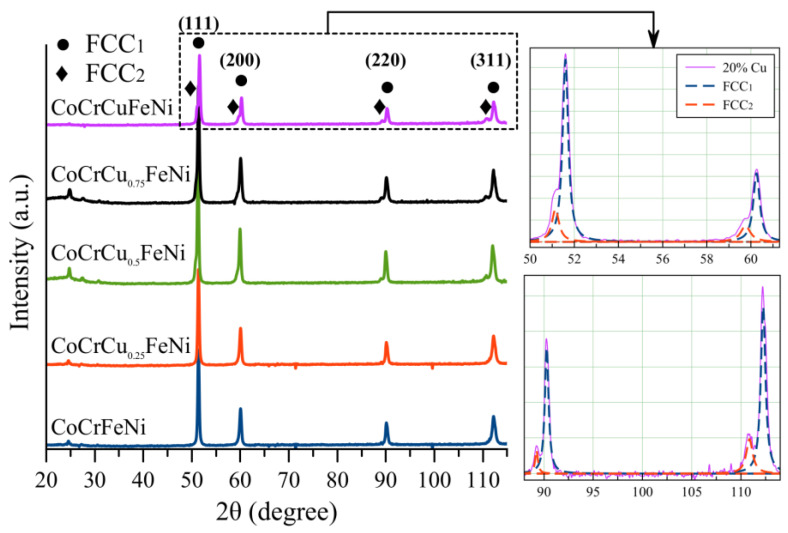
XRD patterns of the CoCrCu_x_FeNi compact samples.

**Figure 5 materials-16-01178-f005:**
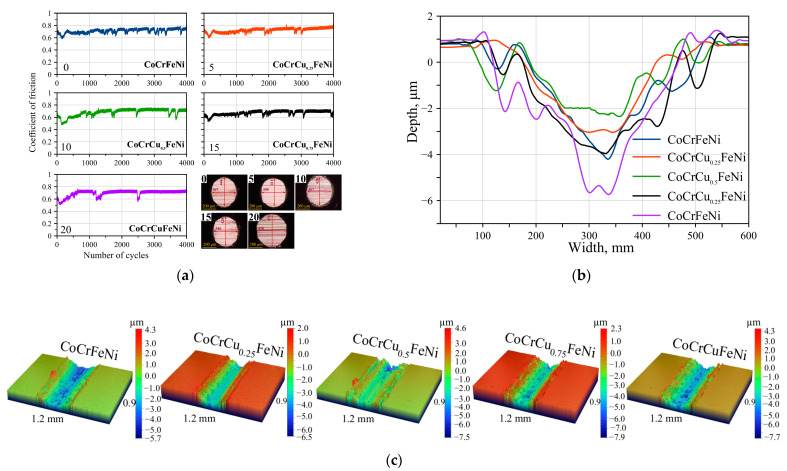
(**a**) The coefficient of friction as a function of the number of cycles (the counterbody wear is shown in the inset); (**b**) wear track profiles for the CoCrCu_x_FeNi samples and (**c**) 3D images of the wear tracks.

**Figure 6 materials-16-01178-f006:**
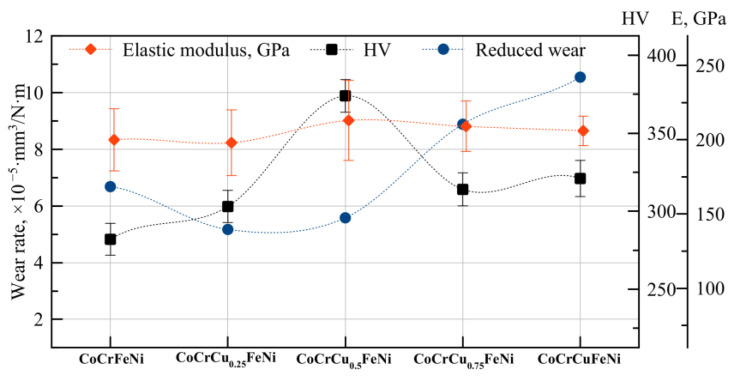
The effect of copper on hardness (HV), the elastic modulus (E, GPa), and wear of CoCrCu_x_FeNi.

**Figure 7 materials-16-01178-f007:**
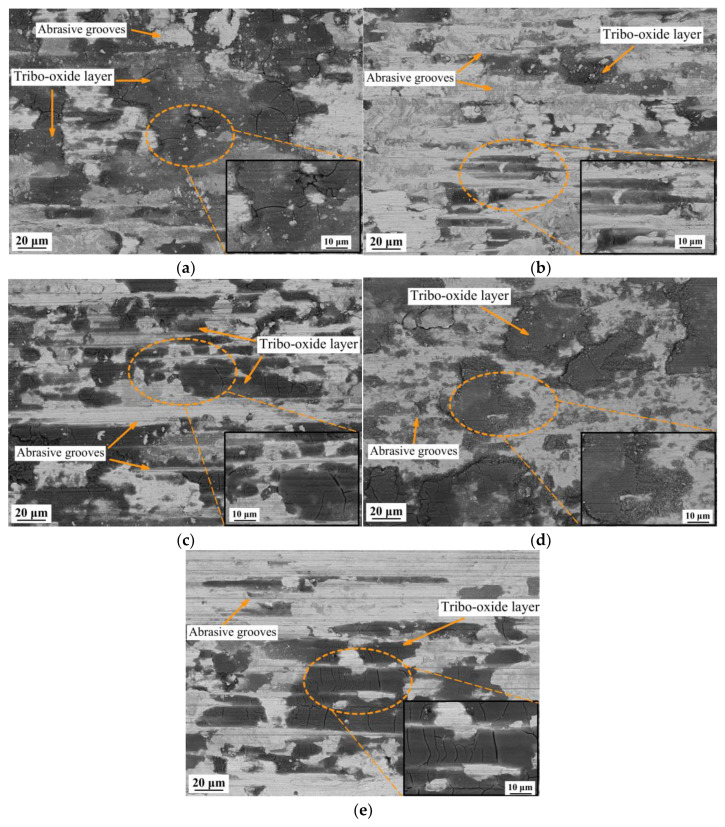
SEM images of the abrasive grooves in the samples: (**a**) CoCrFeNi; (**b**) CoCrCu_0.25_FeNi; (**c**) CoCrCu_0.5_FeNi; (**d**) CoCrCu_0.75_FeNi; (**e**) CoCrCuFeNi.

**Figure 8 materials-16-01178-f008:**
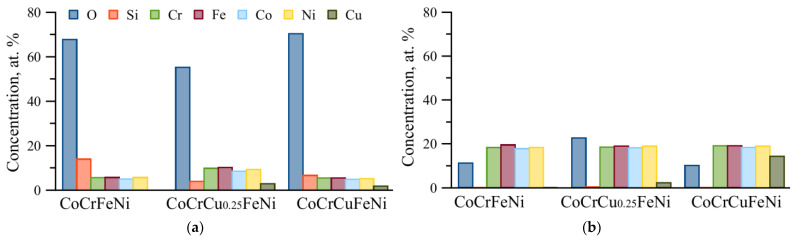
Concentration of elements in the areas where oxidative (**a**) and abrasive wear (**b**) predominate.

**Figure 9 materials-16-01178-f009:**
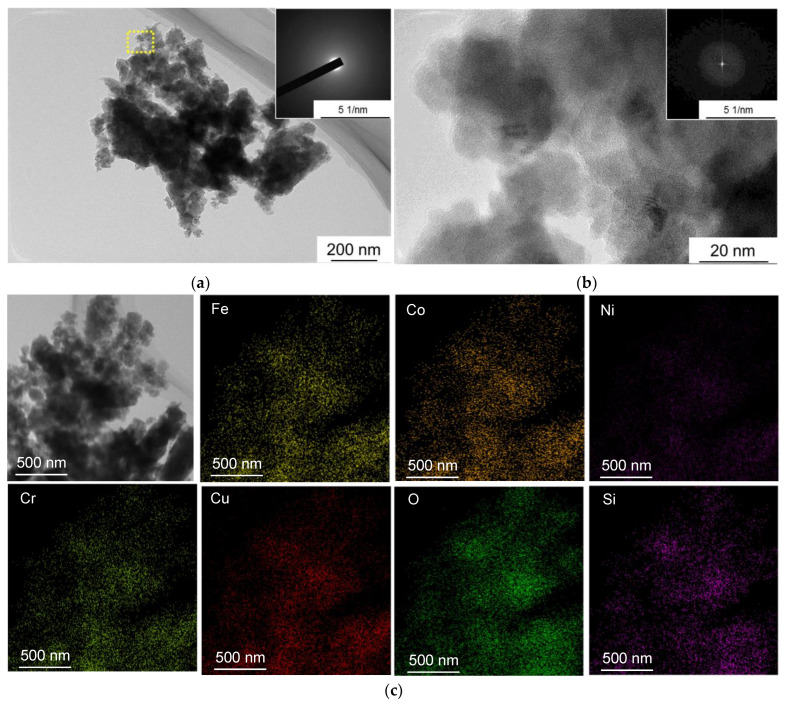
A TEM image of the products in the tribolayer and electron diffraction (**a**); a HR-TEM image of tribolayer particles in the area shown with a yellow rectangle (**b**); the distribution maps of elements in the agglomerate of tribolayer particles (**c**).

**Figure 10 materials-16-01178-f010:**
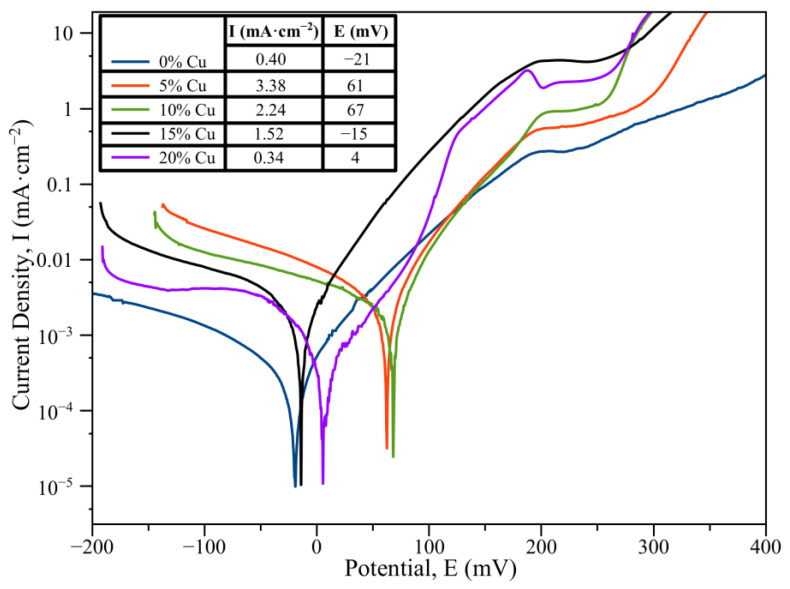
The polarization curves of CoCrCu_x_FeNi in 3.5% NaCl solution.

**Table 1 materials-16-01178-t001:** Tribological properties of HEAs.

Sample	Reduced Wear, 10^−5^·mm^3^/N·m		Coefficient of Friction (COF)		
	Sample	Counterbody	Initial	Maximum	Average
CoCrFeNi	6.68	0.91	0.09	0.81	0.67
CoCrCu_0.25_FeNi	5.17	0.70	0.12	0.83	0.68
CoCrCu_0.5_FeNi	5.58	0.85	0.16	0.85	0.69
CoCrCu_0.75_FeNi	8.88	0.79	0.29	0.81	0.68
CoCrCuFeNi	10.54	1.43	0.41	0.85	0.71

## Data Availability

The raw/processed data required to reproduce these findings cannot be shared at this time as the data also forms part of an ongoing study. Some data can be made available upon request.

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
