# Peer review of "The Effect of Copper on the Microstructure, Wear and Corrosion Resistance of CoCrCuFeNi High-Entropy Alloys Manufactured by Powder Metallurgy"

_materials, 2023, doi:10.3390/ma16031178_

Round 1

Reviewer 1 Report

The authors should explain in more detail the statement: One can see that hardness values are correlated rather tightly with reduced wear, which refers to Figure 6.

Please explain the hardness values and reduced wear values in the points of the CoCrCu0.25FeNi and CoCrCu0.5FeNi copper concentration.

Author Response

We would like to express our thanks to the reviewer for a careful study of our paper and useful recommendations. Obviously, they will improve the quality of our paper. Please find below our point-by-point responses to the comments. All changes in manuscript text were embedded and highlighted in yellow. Revised text contains corrections, made upon reviewer’s recommendations.

Comment 1

The authors should explain in more detail the statement: One can see that hardness values are correlated rather tightly with reduced wear, which refers to Figure 6.

Response

The wording we have chosen is not entirely appropriate. It's more correct to say «One can see that hardness values are correlated rather tightly with wear resistance». Because, as we can see from Figure 6, hardness and reduced wear are inversely proportional.

Comment 2

Please explain the hardness values and reduced wear values in the points of the CoCrCu0.25FeNi and CoCrCu0.5FeNi copper concentration.

Response

The reason for increasing of hardness and decreasing of reduced wear is solid solution strengthening of initial FCC1 solid solution with addition of copper. Maximum solubility of Cu was reached in CoCrCu0.5FeNi – 9 at. % was dissolved in FCC1 phase and a small amount was expended for Cu-rich FCC2 phase precipitation. At higher Cu concentration FCC2/FCC1 phase ratio increases, thus leading to reduction of hardness. We suppose, that FCC2 phase is rather ductile as it contains 90 at. Cu, and it is undesired for wear resistance.

Some minor changes were embedded in the text.

The concentration of Cu-rich FCC2 phase in CoCrCu0.75FeNi and CoCrCuFeNi alloys increases, which is detrimental for hardness and wear resistance.

Reviewer 2 Report

The experimental work is interesting. The phase structure, mechanical, tribological and corrosion properties of as-prepared samples have been studied. In my opinion, this work is interesting and could be accepted after minor revision.

The suggested modifications are listed as follows:

1-The solubility limit of copper in CoCrFeNi solid solution was found to be 9 at.%. What is the main reason for the solubility limit? It should be discussed more. this ref can be a good response (https://doi.org/10.1016/j.ceramint.2019.10.175).Please explain the relation between solubility limit, unwanted phases and etc. in your work.

2-For better investigation of particle size, please add the size distribution in Fig.9(a).

3-Did the synthesized sample ability to be used in medical applications or other environments desired for antibacterial uses?

Author Response

We would like to express our thanks to the reviewer for a careful study of our paper and useful recommendations. Obviously, they will improve the quality of our paper. Please find below our point-by-point responses to the comments. All changes in manuscript text were embedded and highlighted in yellow. Revised text contains corrections, made upon reviewer’s recommendations.

Comment 1

1-The solubility limit of copper in CoCrFeNi solid solution was found to be 9 at.%. What is the main reason for the solubility limit? It should be discussed more. this ref can be a good response (https://doi.org/10.1016/j.ceramint.2019.10.175).Please explain the relation between solubility limit, unwanted phases and etc. in your work.

Response

The description of phase transformation at hot pressing of MA powders was added on Page 6.

The main reason for the limited solubility of Cu in CoCrFeNi solid solution is extremely low mutual solubility of Cu with Co, Cr and Fe. Application of high energy processes, such as mechanical alloying, helps to obtain a solid solution with higher Cu concentration, than it is thermodynamically permissible. Thus, equiatomic MA powder mixture CoCrCuFeNi, obtained in our work, consists of metastable phase, which is exposed to diffusion activated spinodal decomposition at hot pressing. The higher amount of Cu is in CoCrCuxFeNi alloys, the higher concentration of FCC2 phase forms in the alloy.   

Comment 2

2-For better investigation of particle size, please add the size distribution in Fig.9(a).

Response

The precise investigation of particle size distribution for debris after wear tests is challenging. The common techniques, for example based on laser diffraction, are not applicable in this case, because the amount of debris is not enough for analysis.

Low magnification TEM images allow to measure the average particle size of debris by random linear intercept method. We made the corresponding calculations and added the result in the text. The analysis of size distribution using TEM images is challenging as well. The greater part of debris particles are not suitable for analysis due to their overlapping (as it is shown in Figure 9a). Particle size distribution could be measured for outer particles in the agglomerates. But we suppose, that it would not be accurate statistically.

For this reason we have removed the statement about particle size from the text.

Comment 3

3-Did the synthesized sample ability to be used in medical applications or other environments desired for antibacterial uses?

Response

Initially, the synthesized CoCrCuFeNi high-entropy alloys were meant to apply as a binder for diamond cutting or grinding tools, exposed to severe wear during performance. Nevertheless, we did not specify the potential application field in the introduction for this very reason. Wear resistant powder metallurgy manufactured CoCrCuFeNi high-entropy alloys can be used in multiple areas, including medical applications, which was demonstrated in [Kuptsov, K.A.; Antonyuk, M.N.; Sheveyko, A.N.; Bondarev, A.V.; Ignatov, S.G.; Slukin, P.V.; Dwivedi, P.; Fraile, A.;  Polcar, T.; Shtansky, D.V. High-entropy Fe-Cr-Ni-Co-(Cu) coatings produced by vacuum electro-spark deposition for marine and coastal applications. Surface and Coatings Technology. 2023, Volume 453, 129136]

Reviewer 3 Report

As highlighted from the title, this paper presented the effect of copper on the high entropy alloy and their further effect on microstructure, corrosion and wear testing. However, the article has following short comings, which need to be addressed properly:

The effect of copper increment at high is added but the effect at low doping on to the corrosion is missing in abstract.

I think the first paragraph of introduction must address why is there any need of HEA in contrast with the already available alloys along with the targeted application.

Can you provide any reference for lines 63-64?

The authors have opted power metallurgy route for HEA, the other manufacturing techniques needs to be included in a separate paragraph in introduction.

Kindly explain what is 1 and 2 represent with the FCC in abstract and subsequent section for the readers.

Which standard has been followed for the wear testing? Include the specimens dimensions and real pictures of specimens and wear tracks.

Author Response

We would like to express our thanks to the reviewer for a careful study of our paper and useful recommendations. Obviously, they will improve the quality of our paper. Please find below our point-by-point responses to the comments. All changes in manuscript text were embedded and highlighted in yellow. Revised text contains corrections, made upon reviewer’s recommendations.

Comment 1

The effect of copper increment at high is added but the effect at low doping on to the corrosion is missing in abstract.

Response

The description of the effect of low concentrations of copper is added in the abstract.

The corrosion tests of CoCrCuFeNi HEAs in 3.5% NaCl solution had demonstrated that doping the alloy with copper at low concentrations (5-10 %) leads to decreasing of corrosion resistance, possibly due to the formation of undesirable oxide Cu2O along with protective Cr2O3. At high copper concentrations (15-20 %) galvanic corrosion is suppressed due to coarsening of FCC2 grains and thus decreasing the specific contact surface area between the cathode (FCC2) and the anode (FCC1).

Comment 2

I think the first paragraph of introduction must address why is there any need of HEA in contrast with the already available alloys along with the targeted application.

Response

We added the first paragraph with motivation of HEA investigation.

This approach allows to design new alloys with excellent combination of strength, ductility, hardness and wear resistance [5, 6]. The presence of high concentration of uniformly distributed passive oxide-forming elements such as Cr ensures outstanding corrosion resistance [7]. These features of HEAs provide their superiority over traditional steels or iron-based alloys for multi-purpose materials [8-10].

Comment 3

Can you provide any reference for lines 63-64?

Response

The references for the statement in lines 63-64 were added. More detailed description of the results obtained in mentioned works is given below in the paragraph.

Comment 4

The authors have opted powder metallurgy route for HEA, the other manufacturing techniques needs to be included in a separate paragraph in introduction.

Response

We added a paragraph with emphasis on the comparison of powder metallurgy with other methods, involving the formation of liquid phase (arc melting, casting, additive manufacturing).

One of the possible ways to improve the mechanical, tribological properties and corrosion resistance of CoCrFeNi-based HEAs is to apply powder metallurgy techniques. Currently the most commonly used methods for their production (arc melting, casting, additive manufacturing and others) involve the formation of liquid phase, resulting in inhomogeneity, dendritic segregation and formation of eutectic-type structures [26]. Mechanical alloying of powder mixtures followed by hot pressing or spark plasma sintering allows obtaining materials with fine structure and uniform distribution of the elements, despite the limitations or even complete absence of their mutual solubility.

Comment 5

Kindly explain what is 1 and 2 represent with the FCC in abstract and subsequent section for the readers.

Response

We had to introduce the present phases as FCC1 and FCC2 to improve readability. Unlike the most of dual-phase HEAs with phases, having different types of crystal lattice (BCC + FCC, FCC + HCP, etc.), CoCrCuxFeNi alloys are known to consist Cu-lean and Cu-rich phases with the same FCC lattice (cF4/1 Bravais lattice).

We have modified the abstract to emphasize it.

Also we introduced the naming of FCC1 and FCC2 phases at Figure 4 discussion (XRD patterns), where the precipitation of secondary phase is first proven.

Comment 6

Which standard has been followed for the wear testing? Include the specimens dimensions and real pictures of specimens and wear tracks.

Response

We added information about applied standards in the Materials and methods section.

Tribological tests were carried out with a high-precision tribometer (CSM Instruments, Switzerland) according to ASTM G 99-17 and DIN 50324 standards using the “pin-on-disk” scheme upon reciprocating motion.

We added Figure 5c with 3D images of wear tracks for better visualization of the difference between the wear resistance of CoCrCuxFeNi alloys.